# Seebeck and Figure of Merit Enhancement by Rare Earth Doping in Yb_14-x_RE_x_ZnSb_11_ (x = 0.5)

**DOI:** 10.3390/ma12050731

**Published:** 2019-03-03

**Authors:** Elizabeth L. Kunz Wille, Navtej S. Grewal, Sabah K. Bux, Susan M. Kauzlarich

**Affiliations:** 1Department of Chemistry, University of California, Davis, One Shields Avenue, Davis, CA 95616, USA; ewille@ucdavis.edu (E.L.K.W.); nsggrewal@ucdavis.edu (N.S.G.); 2Jet Propulsion Laboratory, California Institute of Technology, 4800 Oak Grove Drive, Pasadena, CA 91109, USA; Sabah.K.Bux@jpl.nasa.gov

**Keywords:** thermoelectric, Seebeck, Yb_14_MnSb_11_, intermetallic, intermediate valence, valence fluctuation

## Abstract

Yb_14_ZnSb_11_ has been of interest for its intermediate valency and possible Kondo designation. It is one of the few transition metal compounds of the Ca_14_AlSb_11_ structure type that show metallic behavior. While the solid solution of Yb_14_Mn_1-x_Zn_x_Sb_11_ shows an improvement in the high temperature figure of merit of about 10% over Yb_14_MnSb_11_, there has been no investigation of optimization of the Zn containing phase. In an effort to expand the possible high temperature p-type thermoelectric materials with this structure type, the rare earth (RE) containing solid solution Yb_14-x_RE_x_ZnSb_11_ (RE = Y, La) was investigated. The substitution of a small amount of 3+ rare earth (RE) for Yb^2+^ was employed as a means of optimizing Yb_14_MnSb_11_ for use as a thermoelectric material. Yb_14_ZnSb_11_ is considered an intermediate valence Kondo system where some percentage of the Yb is formally 3+ and undergoes a reduction to 2+ at ~85 K. The substitution of a 3+ RE element could either replace the Yb^3+^ or add to the total amount of 3+ RE and provides changes to the electronic states. RE = Y, La were chosen as they represent the two extremes in size as substitutions for Yb: a similar and much larger size RE, respectively, compared with Yb^3+^. The composition x = 0.5 was chosen as that is the typical amount of RE element that can be substituted into Yb_14_MnSb_11_. These two new RE containing compositions show a significant improvement in Seebeck while decreasing thermal conductivity. The addition of RE increases the melting point of Yb_14_ZnSb_11_ so that the transport data from 300 K to 1275 K can be collected. The figure of merit is increased five times over that of Yb_14_ZnSb_11_ and provides a *zT* ~0.7 at 1275 K.

## 1. Introduction

Compounds of the Ca_14_AlSb_11_ (14-1-11) structure type have been shown to exhibit high thermoelectric figure of merit, *zT*, at high temperatures [1,2,3,4]. While Yb_14_MnSb_11_ and Yb_14_MgSb_11_ members of this group have been high achievers in this area [5,6], the more metallic Yb_14_ZnSb_11_ has never been considered a good thermoelectric material because of its low Seebeck coefficient (*α*) and, therefore, low *zT*, as it scales with *α*^2^ [7,8]. However, the low electrical resistivity that it possesses is an attractive feature, and prior work sought to tap into this by forming a solid solution of Zn with Mn, which resulted in improved *zT* compared with Yb_14_MnSb_11_ [9]. Yb_14_ZnSb_11_ has a smaller unit cell and possesses a lower decomposition temperature than those of its Mn and Mg counterparts; the latter property further dashing hopes for its use in high temperature TE devices. Yb_14_ZnSb_11_ is unique amongst the members of the 14-1-11 family in that it exhibits Curie-Weiss behavior equivalent to about 0.75 Yb^3+^ from 300 K to 100 K and a broad maximum in magnetic susceptibility at around 85 K that drops as temperature is lowered, followed by a sharp increase at 20 K. The fact that there is not a simple integral amount of Yb^3+^ is consistent with an “intermediate valence”. The broad maximum is interpreted as a fluctuation between the *f^13^* (3+) and *f^14^* (2+) electronic configurations of Yb, while the low temperature increase in susceptibility is attributed to free Yb^3+^ impurities. Intermediate valence is observed in some of the rare earth elements, such as Ce, Eu, and Yb [10]. The resulting change in valence corresponds to the effective nuclear charge and then, ultimately, to a change in lattice parameters [7]. The valence fluctuation in Yb_14_ZnSb_11_ is the shift from the small percentage of Yb^3+^ states at high temperature to all Yb^2+^ at a low temperature. A Curie–Weiss fit of the paramagnetic region above 150 K yields a *µ_eff_* of 3.8 µB, which corresponds to the presence of approximately 0.8 Yb^3+^ per formula unit [7]. The existence of 0.75 Yb^3+^ in this compound makes Yb_14_ZnSb_11_ close to a valence precise Zintl formula, but the low resistivity and intermediate valence of Yb distinguish it from this simplistic interpretation of bonding. Recently, magnetic susceptibility measurements of the Mg compound were reported and are consistent with a similar amount of Yb^3+^, but there is no evidence for intermediate valency [11]. Yb_14_MnSb_11_ contains only Yb^2+^, confirmed by X-ray photoelectron spectroscopy (XPS) and X-ray magnetic circular dichroism (XMCD) and neutron measurements [8,12].

Figure 1 shows a view of the unit cell of Yb_14_ZnSb_11_ along the *c* axis. This compound is considered as a Zintl phase with the approximate formula of 13Yb^2+^ + ~1Yb^3+^ +ZnSb_4_^10−^ + Sb_3_^7−^ + 4Sb^3−^ [7]. There are four Yb crystallographic sites in the structure, but there is no direct evidence from the structure concerning site preference for the Yb^3+^ cation, although Yb_14_ZnSb_11_ does have the smallest lattice parameters within this family of compounds. While the valence precise Zintl phase of Yb_14_AlSb_11_ has been shown to have semiconducting electrical transport properties, Yb_14_ZnSb_11_ shows the lowest resistance of compounds of this structure type published to date. The crystal structures of Yb_14_ZnSb_11_ and Ca_14_ZnSb_11_ were reported with defects or interstitial atoms; Yb_14_ZnSb_11_ contains a slight deficiency on the Zn site and Ca_14_ZnSb_11_ is purported to contain interstitial Sb [8,13]. The low resistance of Yb_14_ZnSb_11_ is attributed to either the intermediate valence of Yb or to the defects in the structure [7].

In Yb_14_MnSb_11_, the substitution of 3+ rare earth (RE) cations for Yb cations in small amounts (x < 0.5) has been successful in improving *zT* and, in addition, has been shown in some cases to decrease the high temperature sublimation (as is the case for RE = La) [14]. A slight reduction in carrier concentration from the substitution of the RE helps to boost α and, in turn, *zT*. In all attempts, no more than x ~0.7 was found to incorporate into the structure of single crystals of Yb_14-x_RE_x_MnSb_11_ solution grown in Sn flux [14,15,16,17]. The isostructural Ca_14-x_RE_x_MnSb_11_ grown in Pb flux is reported to exhibit a limit of x = 1 [18]. It is not clear if the differences in substitution for the two different parent phases, Yb_14_MnSb_11_ versus Ca_14_MnSb_11_, is due to the different flux employed or electronic and size effects.

In an effort to further expand our investigation of the effect of RE^3+^ on the transport properties of this structure type, the solid solution, Yb_14-x_RE_x_ZnSb_11_ (RE = Y and La), was investigated. The solid solutions were made via a stoichiometric metallurgical approach and the samples condensed into fully dense pellets for measurement. Seebeck, electrical and thermal transport, and Hall measurements are reported.

## 2. Materials and Methods

### 2.1. Synthesis

Samples of Yb_13.5_RE_0.5_ZnSb_11_ were synthesized by combining Yb filings, Sb shot, Zn shot (100% excess), and RE filings with a total mass of 8 g in a SPEX 55ml tungsten carbide canister with one large and two small tungsten carbide balls. Work was performed in an argon filled drybox and both RE and Yb were brushed with a designated wire brush prior to filing to remove any oxide on the surface. Samples were milled using a SPEX 8000M mixer mill (SPEX, 65 Liberty Street, Metuchen, NJ, USA) for a total of 1 h and 30 min, with 15 min of rest time between 30 min milling intervals, and a scrape down inside the drybox after 1 h of milling time. Samples were sealed in 13 cm long Nb tubes, arc melted shut under Ar, and sealed in quartz under vacuum. The samples were annealed for 96 h at 900 °C in a box furnace. Zn was used in 100% excess in an effort to prevent formation of Yb_11_Sb_10_. Samples made with a stoichiometric amount of Zn contained this side phase as 20% or larger composition, indicating some loss of Zn during the ball milling or annealing stage. 

### 2.2. Consolidation of Powder

Annealed powder samples were made into dense pellets for measurement via a spark plasma sintering (SPS) Dr. Sinter Lab SPS-211LX unit (Fuji Electric Industrial Co., Ltd, 6-2-22 Fujimi, Tsurugashima, Saitama, Japan). In an argon drybox, the annealed powder was ground in an agate mortar and pestle and passed through a 200 mesh stainless steel sieve and loaded between multiple thin graphite foil spacers in a 12.7 mm inner diameter high-density graphite die. Sintering was performed under dynamic vacuum and with a starting sample pressure of 20 MPa. The temperature was ramped from 20 °C to 750 °C over four minutes, then to 800 °C in one minute to avoid temperature overshoot. The pressure was slowly and steadily increased to 63 MPa during the temperature range 700–800 °C (about 1.5 min). Then, 800 °C and 63 MPa were held constant for 15 min, after which the sintering process was ended, and pressure/temperature released. Pressed pellets were typically 2 g in size and were cut circumferentially into two disks using a Buehler diamond saw to allow for one to be pulverized for use in characterization via powder X-ray diffraction. The other pellet was saved for properties measurements. The pellet densities obtained through this sintering profile were greater than 96% of the theoretical densities for each compound. 

### 2.3. Electron Microprobe Analysis and Wavelength Dispersive Spectroscopy

After measurement of TE properties, small pieces of pellets were mounted in epoxy and polished using grits sizes down to 0.01 µm. Care was taken to prevent oxidation of these polished sample pucks and, after preparation, they were stored under dynamic vacuum and transported triple-bagged in argon atmosphere. Prior to their measurement, the pucks were carbon coated to prevent charging. Samples were analyzed using a Cameca SX100 electron microprobe (CAMECA Instruments, Inc., 5470 Nobel Drive, Madison, WI, USA) with five wavelength dispersive X-ray spectrometers, operated at 15 kV accelerating potential and beam current of 20 nA. A polished single crystal of Yb_14_MnSb_11_ was used as wavelength dispersive X-ray spectroscopy (WDS) standard for Yb. Zn and Sb metal, LaPO_4_, and yttrium aluminum garnet (YAG) crystals were used as WDS standards for Zn and Sb, La and Y, respectively. The composition of each sample was determined by calculating the average and standard deviation of 15 data points of the main phase and 5 data points of the side phase randomly spaced through the sample.

### 2.4. Powder X-Ray Diffraction

Powder X-ray diffraction (PXRD) data were collected on each sample after furnace annealing and after consolidation in the SPS. Samples were ground into a fine powder by mortar and pestle in an Ar drybox and plated with ethanol to obtain a uniform, thin spread onto a zero background holder on a Bruker D8 Advance Eco Diffractometer (BRUKER AXS, Inc., 5465 East Cheryl Parkway, Madison, WI, USA) operated at 40 kV and 25 mA utilizing Ni filtered Cu Kα radiation with the knife-edge attachment. Data were collected from 20° to 80° *2θ* with a step size of 0.19° at 1.5 s. Data were converted from .raw to .gsas using powdll and analyzed via Rietveld refinement using General Structure Analysis System, GSAS-II [19,20]. The GSAS-II instrument parameter file used in refinement was generated from a similarly-prepared LaB_6_ standard. Lattice parameters of the RE phases were obtained from refinement of a 14-1-11 phase modelled from published Crystallographic Information File (CIF) of Yb_14_ZnSb_11_. 

### 2.5. Electrical Resistivity, Hall Effect, and Seebeck Coefficient

The electrical resistivity (*ρ*) and Hall coefficient were measured simultaneously from 300 K to 1275 K on a home-built instrument under dynamic vacuum. Resistivity was measured via the van der Pauw technique using a current of 100 mA; Hall was measured under a forward and reverse magnetic field of about 7500 G. The carrier concentration (*n*) was calculated from *n* = 1/*R_H_e**,* where *R_H_* is the measured Hall coefficient and *e* the elementary charge. The hall factor was assumed to be 1 [21]. The Seebeck coefficient (*α*) was measured using a home-built instrument with graphite heater using W/Nb thermocouples and the temperature differential generated by light pulse. The resultant resistivity and Seebeck data from the heating up measurements were each fitted to a six-order polynomial function for the calculation of *zT*.

### 2.6. Thermal Conductivity

Thermal diffusivity (*D_t_*) data were collected from 300 K to 1275 K using a Netzsch LFA-457 laser flash unit (Netzsch Instruments North America, 129 Middlesex Turnpike Burlington, MA, USA). Then, 12.7 mm diameter pellet samples were polished to obtain parallel top and bottom surfaces and overall thickness less than 1.2 mm, and were then coated in graphite. The measurement was performed under dynamic vacuum and with three data points per temperature step. The Cowan + pulse correction fit of the detected signal was employed through the Netzsch software to obtain values of thermal diffusivity, which were then averaged for each temperature step. Thermal conductivity was calculated via *κ* = *D_t_* × *ρ* × *Cp*, where *ρ* = density and *Cp* = heat capacity as a function of temperature [21]. Room-temperature density was measured geometrically and high-temperature density was estimated using thermal expansion data from previous study on Yb_14_MnSb_11_ [22]. The previously reported experimentally-determined *Cp* values for Yb_14_MnSb_11_ were used as an estimate for these compounds correcting for mass [23].

## 3. Results

The two compounds, Yb_13.5_RE_0.5_ZnSb_11_ (RE = Y, La), were prepared with excess Zn in order to prevent the formation of the unwanted side phase Yb_11_Sb_10_. We have shown in previous publications that the highest temperature properties of compounds of this structure type are compromised once Yb_11_Sb_10_ forms [24]. Synthesis of phase pure Yb_14_MgSb_11_ requires 20% excess Mg; this requirement is attributed to the high vapor pressure of Mg at the reaction temperature. Zn has a slightly higher vapor pressure than that of Mg at the reaction and sintering temperatures, highlighting the need for excess [25]. The samples were prepared by balling the elements, sealing the fine powder into niobium tubes, and heat treating at 900 ℃. The product was then pressed into a dense pellet via spark plasma sintering (SPS). 

Yttrium and lanthanum rare earth elements were chosen for this study because of their sizes. Y^3+^ (0.900 Å) is closest in size to Yb^3+^ (0.868 Å), while La^3+^ (1.032 Å) represents the largest of the 3+ RE cations [26]. As previously mentioned, there are four crystallographic sites for the Yb cation in Yb_14_ZnSb_11_ coordinated by antimony with various sized polyhedral volumes. The site specificity of various rare earth elements has been shown to be correlated with size in studies of Yb_14-x_RE_x_MnSb_11_. Early RE cations with larger ionic radius, such as La, were shown to preferentially substitute on the Yb2 and Yb4 sites, while RE of smaller ionic radius such as Y substitutes on all of the Yb sites [15,16]. While it is expected that carrier concentration plays the largest role in controlling the transport properties, the RE site selectivity has been indicated as important for subtle differences in thermoelectric properties across the series, Yb_14-x_RE_x_MnSb_11_ [2,27].

Electron microprobe X-ray maps of the dense pellets (Figure 2) show that the samples have a good distribution of the elements and that there is excess Zn at the grain boundaries. Appendix A shows the microprobe backscatter electron images of Yb_13.5_Y_0.5_ZnSb_11_ and Yb_13.5_La_0.5_ZnSb_11_. Wavelength dispersive X-ray spectroscopy of the samples show two phases: a main phase (Yb_13.5_RE_0.5_ZnSb_11_) and side phase (Yb_1.95_RE_0.0.5_Zn_0.8_Sb_2)_, tabulated in Table 1. While the main phase was loaded as Yb_13.5_RE_0.5_ZnSb_11_, the analysis shows that when RE = Y, the amount incorporated is slightly less. Whereas for RE = La, it is in good agreement, and the Zn is slightly deficient in both samples, giving rise to the stoichiometries Yb_13.7_Y_0.35_Zn_0.85_Sb_11_ and Yb_13.7_La_0.48_Zn_0.91_Sb_11_. The WDS data were normalized to 11 Sb and while that provides a slightly high Yb + RE content, it is within error consistent with the stoichiometry of 14-1-11, with deficiencies of Zn.

The WDS of the side phase provides a formula that is consistent as a solid solution of RE and ‘Yb_2_ZnSb_2_’ with slight deficiency of Zn. The phase Yb_2_ZnSb_2_ is as of yet unreported, and the obvious possible analog, Ca_2_ZnSb_2_, is also not a reported phase. Rietveld refinement of powder X-ray diffraction data for each of these samples included the phases Yb_14_ZnSb_11_ and Yb_2_O_3_; small unidentified peaks were present after refinement attributed to this side phase. There are reports of the Eu_2_ZnSb_2_ and Sr_2_ZnSb_2_ phase that crystallize in the *P6_3_/mmc* space group [28]. Attempts to unambiguously identify these peaks with the appropriately scaled lattice parameters of known 2-1-2 structure types employing the elements Yb, Zn, and Sb were unsuccessful. Figure 3 contains a zoomed-in overlay of the PXRD data from Yb_13.5_RE_0.5_ZnSb_11_, with the unidentified peaks marked. Unit cell parameters of Yb_13.5_RE_0.5_ZnSb_11_ obtained from the refinement are listed in Table 2. Representative PXRD data are provided in Appendix A. Because the two pellets show similar amounts of this unknown phase and the majority of the phase is the Yb_13.5_RE_0.5_ZnSb_11_, measurements of the thermoelectric and transport properties will provide some insight into the effects of the RE solid solution.

Figure 4 contains the plots of the electrical resistivity, Seebeck, and thermal conductivity of the samples. Both heating and cooling data sets for resistivity and Seebeck are provided in Appendix A. As mentioned previously, Yb_14_ZnSb_11_ has low electrical resistivity, similar to that seen in many intermediate Yb valence compounds, and magnetic susceptibility is consistent with the presence of about 0.75 Yb^3+^ [7]. This mixture of Yb^2+^ and Yb^3+^ can be more exotic and can be described as an intermediate valence state. Yb containing intermetallics can show this effect when the nearly degenerate 4 *f*^13^ and 4*f*^14^ electron levels are close to the *s-d* band, favoring an intermediate valence state. Rare earth ions in this state fluctuate between two 4*f* electronic configurations competing for stability. With doping, the hybridization strength of the *f*-electrons with the conduction electrons can change, resulting in a change in the effective mass and thereby the associated transport properties [29,30]. The electrical resistivity of Yb_13.5_RE_0.5_ZnSb_11_ shows a significant increase at temperatures above 500 K over Yb_14_ZnSb_11_ for both samples. In the Zintl electron counting scenario, RE^3+^ adds one electron to the p-type Yb_14_ZnSb_11_ and is thus expected to reduce the carrier concentration and thereby the electrical resistivity. Consistent with the slightly higher amount of RE in the sample, the RE = La sample shows a slightly higher resistivity value. Consistent with the electrical resistivity, the thermal conductivities of the samples are reduced from that of Yb_14_ZnSb_11_. Lattice thermal conductivity is provided in Appendix A. This is attributed to both the loss of electrical conduction at a high temperature and, from point defect scattering, of the solid solution. There is a decrease in thermal conductivity even at 300 K compared with Yb_14_ZnSb_11_. The Seebeck coefficient shows a remarkable increase over that of Yb_14_ZnSb_11_ for the entire temperature range, with the RE = La sample showing a slightly higher Seebeck at the highest temperatures, consistent with the slightly larger amount of RE cation. This suggests that the effect of the RE^3+^ is to change the hybridization of Yb/RE, thereby leading to a change in bands that are important for the high temperature behavior. 

Figure 5 shows the Hall mobility and Hall carrier concentration of the RE solid solutions. The RE element was substituted with the goal of reducing the carrier concentration and making this compound a better thermoelectric material. The carrier concentrations of both RE samples are lower than that of Yb_14_ZnSb_11_, which shows conductive electrical resistivity at low temperatures and presumably has a high carrier concentration. Typically, for transition metal containing compounds with the formula Yb_14_MSb_11_, where Yb is considered to be all Yb^2+^ and M = M^2+^, the carrier concentration is equivalent to one hole in the unit cell volume. Therefore, the addition of an RE^3+^ cation provides one additional electron to reduce the p-type carrier concentration. In this example, considering the effect of the RE^3+^ cation is complicated because this compound has both Yb^2+^ and Yb^3+^ at room temperature. If the Y^3+^ or La^3+^ cation does not simply substitute for Yb^3+^ in Yb_14_ZnSb_11_ and instead substitutes for Yb^2+^, it would contribute an extra 0.5 electron per formula unit (or 0.35 in the case of Y). Calculating the carrier concentration, it would contribute approximately 6.6 × 10^20^ carriers/cm^3^. This would indicate that at room temperature, the carrier concentration of Yb_14_ZnSb_11_ should be 1.3 × 10^21^ cm^−3^, a value close to the highest room temperature concentrations obtained for Yb_14_MnSb_11_, which is much less metallic than Yb_14_ZnSb_11_. In a similar system, Yb_14-x_La_x_MnSb_11_ (x = 0.4, 0.7) was found to have a reduction in room temperature carrier concentration from that of Yb_14_MnSb_11_ (1.1 to 1.3 × 10^21^ cm^−3^), which closely corresponded with the amount of La added, 6 × 10^20^ cm^−3^ and 4 × 10^20^ cm^−3^ for 0.4 La and 0.7 La, respectively [14,17]. Therefore, these results suggest that RE^3+^ is substituting for Yb^3+^ in Yb_14_ZnSb_11_ and that once the Yb^3+^ is no longer a species in the structure, the metallic conduction is no longer viable. Because neither Y nor La have filled *f* electrons, it is possible that a hybridized band from Yb^3+^ is responsible for the low electrical conduction in Yb_14_ZnSb_11_. Considering the reductions in carrier concentrations from the Y^3+^ and La^3+^ substitutions, the large increase in Seebeck is consistent.

Figure 6 shows the *zT* for the Yb_13.5_RE_0.5_ZnSb_11_ (RE = Y, La) compounds compared with the *zT* of Yb_14_ZnSb_11_. The properties of Yb_14_ZnSb_11_ were only measured up to 900 K because of the stability of the compound. With the addition of the RE, the Yb_13.5_RE_0.5_ZnSb_11_ (RE = Y, La) compounds are stable to 1275 K. This is a side benefit of RE^3+^ incorporation that has been also noted for Yb_14_MnSb_11_, where the melting point is increased and sublimation vapor pressure is decreased depending upon the identification and amount of rare earth ion incorporation [31].

Figure 7 contains Pisarenko plots at 400 K, 800 K, and 1200 K that were generated using a single parabolic band (SPB) model. The parameters used to generate these plots are provided in Table 3. The effective mass values generated for this model at 1200 K for both RE = La, Y are significantly larger than those generated at 400 and 800 K. These parameters indicate that modelling Yb_13.5_RE_0.5_ZnSb_11_ as a single parabolic band is insufficient and that the band(s) change from light to heavy with temperature [32]. This is supported by the reduction in carrier concentration that these samples exhibit with only a small donation of 0.5 or less extra e^-^ density per formula unit. These plots suggest that the carrier concentration could be further reduced to obtain peak *zT*.

## 4. Conclusions

The addition of the rare earths, Y and La, to the Yb_14_ZnSb_11_ system has a profound but complex effect on the carrier concentration and presumably the density of states (DOS) as a function of temperature. The large improvement in *zT* observed in the Yb_13.5_RE_0.5_ZnSb_11_ (RE = Y, La) samples over Yb_14_ZnSb_11_ is unexpected because these RE^3+^ ions are simply replacing Yb^3+^. These remarkable results suggest that better modeling/theoretical understanding of complex systems is important to further advance the field. Renewed interest in the nuanced system of Yb_14_ZnSb_11_ may lead to a more complete understanding of the electronic and structural factors affecting the 14-1-11 compounds and aid in the future design of optimized materials. Further improvement to the *zT* of these compounds might be achieved by reducing carrier concentration further by means of increasing x or by substitution of Ca on the Yb site or Al on the Zn site. Yb_14-x_Ca_x_MnSb_11_ and Yb_14_Mn_1-x_Al_x_Sb_11_ solid solutions show reduced carrier concentration with increasing x and higher zT’s than Yb_14_MnSb_11_. While x has been shown to be limited in the case of Yb_14-x_RE_x_MnSb_11_ to x ~0.5, it might be possible to increase x to 1 for the Zn 14-1-11 phase, as is the case for Ca_14-x_RE_x_MnSb_11_. Overall, these results for Yb_13.5_RE_0.5_ZnSb_11_ (RE = Y, La) suggest that there is significant room for improvement of zT with new compositions of this structure type.

## Figures and Tables

**Figure 1 materials-12-00731-f001:**
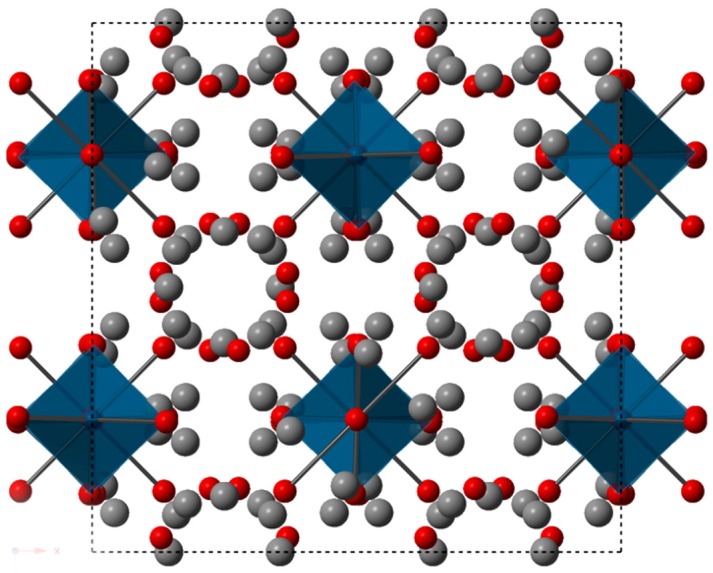
A view of the structure of Yb_14_ZnSb_11_ down the c axis. Yb atoms are grey, Sb atoms are in red, and the blue tetrahedra are the ZnSb_4_ units.

**Figure 2 materials-12-00731-f002:**
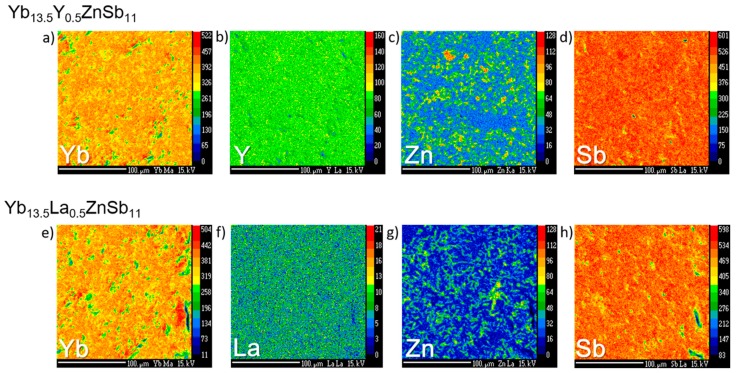
Electron microprobe X-ray maps of pelleted samples of Yb_13.5_Y_0.5_ZnSb_11_ (**a**–**d**) and Yb_13.5_La_0.5_ZnSb_11_ (**e**–**h**).

**Figure 3 materials-12-00731-f003:**
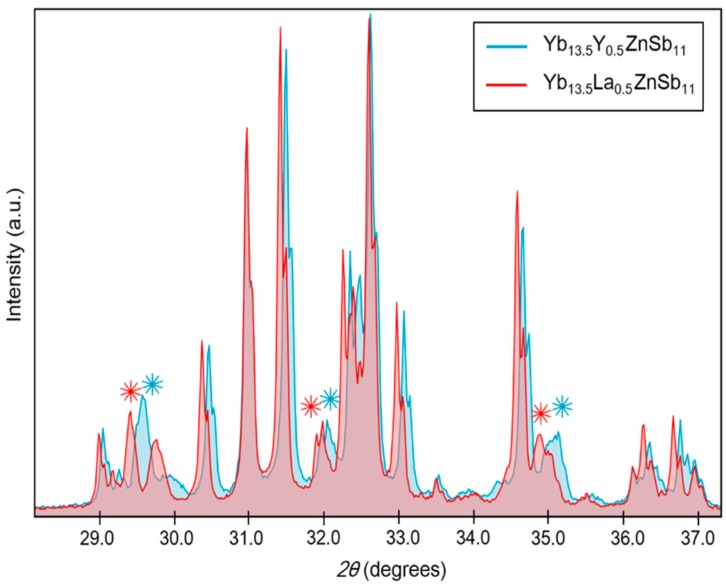
Powder X-ray diffraction (PXRD) patterns of Yb_13.5_Y_0.5_ZnSb_11_ (filled in blue) and Yb_13.5_La_0.5_ZnSb_11_ (filled in red) from 28° to 37° *2θ*. Unidentified peaks in each pattern are marked by asterisks in respective colors.

**Figure 4 materials-12-00731-f004:**
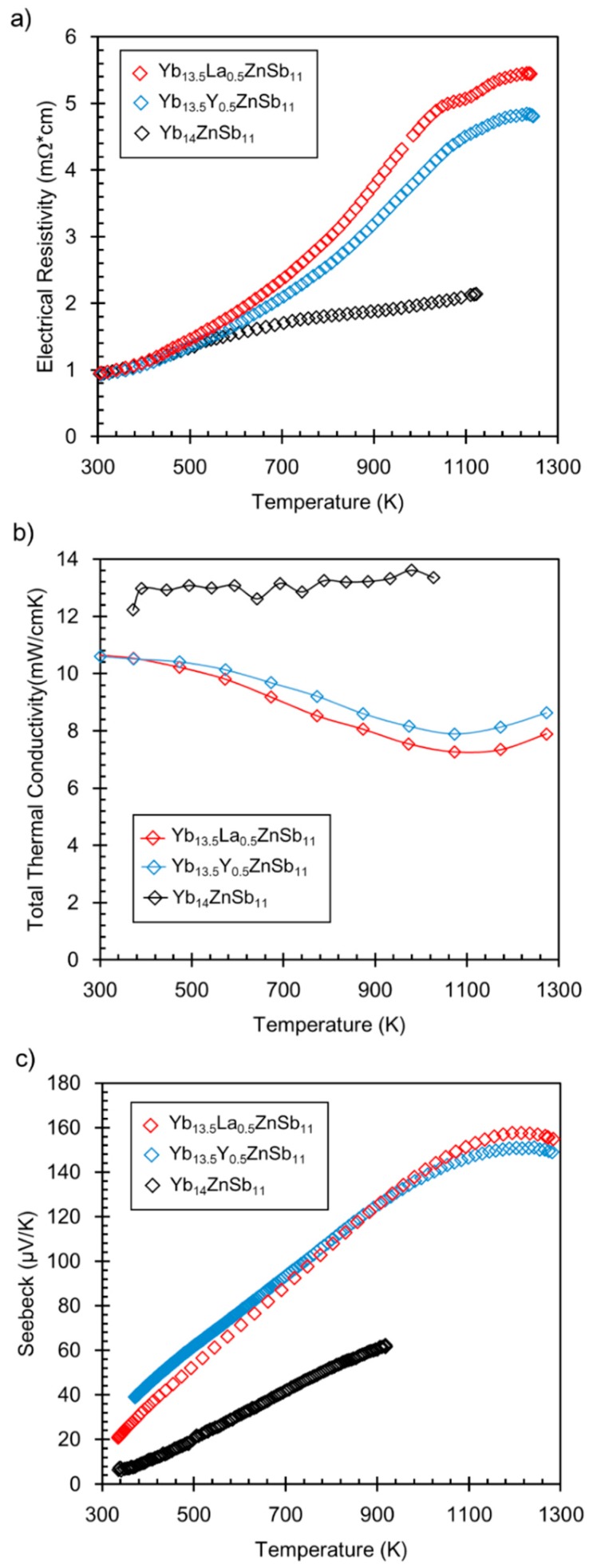
(**a**) Electrical resistivity, (**b**) thermal conductivity, and (**c**) Seebeck for the Yb_13.5_RE_0.5_ZnSb_11_ (RE = Y, La) compounds plotted against those of Yb_14_ZnSb_11_ (Yb_14_ZnSb_11_ data from the work of [9]).

**Figure 5 materials-12-00731-f005:**
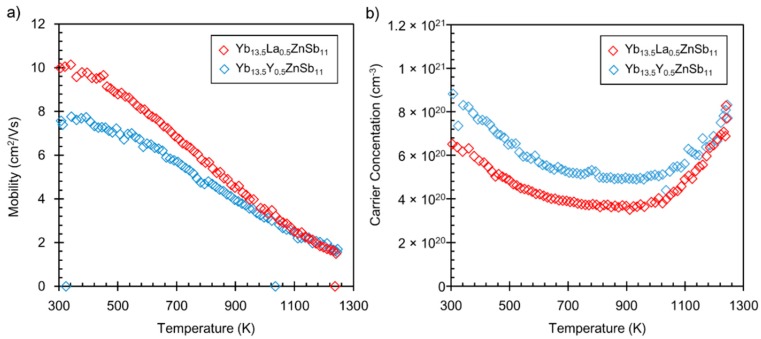
(**a**) Mobility and (**b**) carrier concentration of the Yb_13.5_RE_0.5_ZnSb_11_ (rare earth (RE) = Y, La) compounds.

**Figure 6 materials-12-00731-f006:**
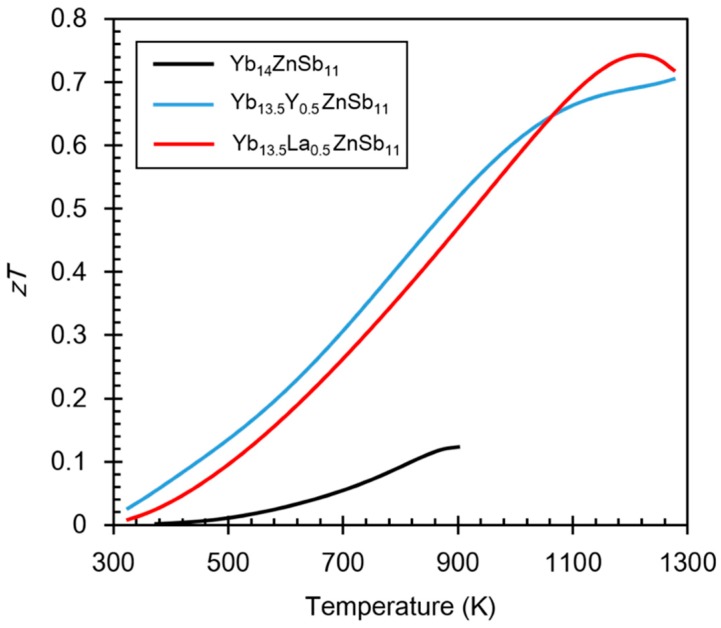
Calculated *zT* for the Yb_13.5_RE_0.5_ZnSb_11_ (RE = Y, La) compounds compared with that of Yb_14_ZnSb_11_ (data from the work of [9]).

**Figure 7 materials-12-00731-f007:**
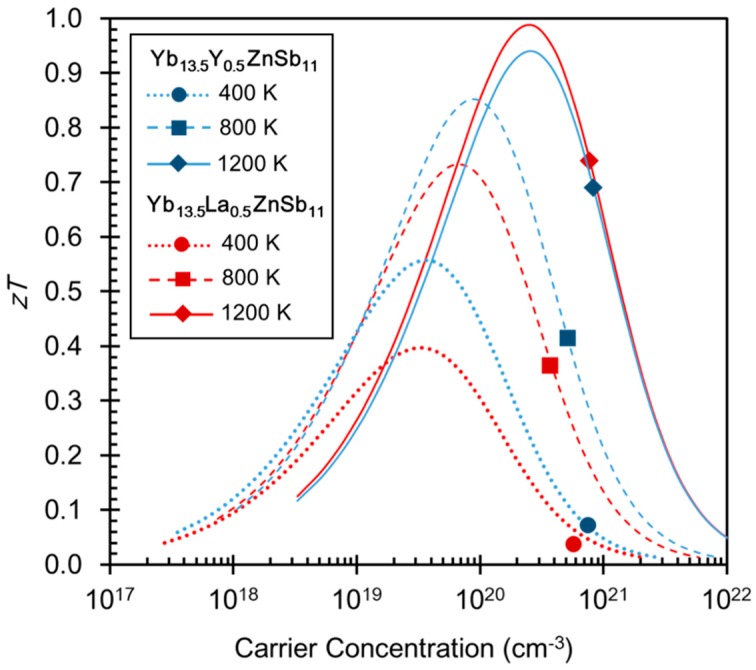
Pisarenko plots for the Yb_13.5_RE_0.5_ZnSb_11_ (RE = Y, La) compounds calculated at 400, 800, and 1200 K.

**Table 1 materials-12-00731-t001:** Wavelength dispersive X-ray spectroscopy (WDS) stoichiometry from pelleted samples from an average of 15 data points (main phase) and an average of 5 points (side phase). RE—rare earth.

	As Loaded	Yb	RE	Zn	Sb
**Main Phase**	Yb_13.5_Y_0.5_ZnSb_11_	13.7(2)	0.35(1)	0.85(5)	11.0(1)
	Yb_13.5_La_0.5_ZnSb_11_	13.7(2)	0.48(5)	0.91(5)	11.0(1)
**Secondary Phase**	Yb_13.5_Y_0.5_ZnSb_11_	1.96(2)	0.04(1)	0.78(2)	2.00(2)
	Yb_13.5_La_0.5_ZnSb_11_	1.95(2)	0.08(1)	0.79(2)	2.00(2)

**Table 2 materials-12-00731-t002:** Lattice parameters as determined by refinement of powder X-ray diffraction (PXRD) data using GSAS II.

As Loaded	a (Å)	c (Å)	V (Å^3^)	wR (Overall)	RF^2^/RF (14-1-11 Phase)
Yb_13.5_Y_0.5_ZnSb_11_	16.5939(4)	21.9309(7)	6038.9(3)	20.812%	14.122%/9.616%
Yb_13.5_La_0.5_ZnSb_11_	16.6412(4)	21.9188(6)	6070.0(3)	19.914%	12.328%/8.316%

**Table 3 materials-12-00731-t003:** Values used in generating the Pisarenko plots shown in Figure 7.

Sample	*T* (K)	*m** (m_0_)	*µ_0_* (cm^2^/V∙s)	*κ_L_* (mW/cm∙K)
**Yb_13.5_Y_0.5_ZnSb_11_**	400	1.40	18.95	2.4
	800	1.47	7.61	3.2
	1200	2.07	2.11	3.8
**Yb_13.5_La_0.5_ZnSb_11_**	400	1.18	24.44	3.4
	800	1.15	9.3	3.4
	1200	2.07	2.08	3.5

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
