# Peer review of "Seebeck and Figure of Merit Enhancement by Rare Earth Doping in Yb14-xRExZnSb11 (x = 0.5)"

_materials, 2019, doi:10.3390/ma12050731_

Round 1

Reviewer 1 Report

The manuscript "Seebeck and Figure of Merit Enhancement by Rare 2 Earth Doping in Yb14-xRExZnSb11 (x = 0.5)" describes a successful way for improvement of promising thermoelectric materials. The results are good and the methods appropriate; however, some improvements are necessary.

In the abstract it would make more sense to motivate research based on more related Yb14-xRExMnSb11 than on Yb14MnxZn1-xSb11. The choice of Y as extreme of size for RE is at least strange. Even ecluding Sc there are a few other RE smaller than Y.

Introduction is somewhat too short and element restricted. Please add at least a short paragraph with the topic overview for general readers showing the role of different RE/alkaline-earths and transition metals in this structure type.

L38-39: The authors are expected to be more accurate in definitions - what is meant by intemediate valence of Yb3+? Zn compound contains both Yb2+ and Yb3+ same as the Mg containing one, but only Zn compound is considered unique. Why?

L50-51 "The low resistance is 50 attributed to either the intermediate valency of Yb or to the defects in the structure"  Electron precise Zintl phase is generally expected to have lower conductivity. How do these facts coexist?

L144-151: The statement about 4 crystallographic Yb and Yb2/Yb4 sites is practically completely duplicated. 

L161: Total occupation of the RE sites exceed 14, while Zn is deficient. Would this mean RE partially occupies Zn site? No comment is provided

L167-169 The caption contains unrelated infomation regarding PXRD

L171 Please change deficient to deficiency

Figure 3: How do you explain different intensities of the peak around 29.8-30 for blue and red plots?

L210: "Figure 4. (a) Electrical resistivity, (b) thermal conductivity (c) and Seebeck"

Please change to "Figure 4. (a) Electrical resistivity, (b) thermal conductivity  and (c) Seebeck"

Author Response

The manuscript "Seebeck and Figure of Merit Enhancement by Rare 2 Earth

Doping in Yb14-xRExZnSb11 (x = 0.5)" describes a successful way for

improvement of promising thermoelectric materials. The results are good and

the methods appropriate; however, some improvements are necessary.

In the abstract it would make more sense to motivate research based on

more related Yb14-xRExMnSb11 than on Yb14MnxZn1-xSb11. The choice of

Y as extreme of size for RE is at least strange. Even ecluding Sc there are a

few other RE smaller than Y.

Response: The wording has been changed in the abstract to more clearly convey the motivation to the reader.

Introduction is somewhat too short and element restricted. Please add at

least a short paragraph with the topic overview for general readers showing

the role of different RE/alkaline-earths and transition metals in this structure

type.

Response: Some information about the Ca/RE system has been added

L38-39: The authors are expected to be more accurate in definitions - what is

meant by intemediate valence of Yb3+? Zn compound contains both Yb2+

and Yb3+ same as the Mg containing one, but only Zn compound is

considered unique. Why?

Response: The Zn compound exhibits Yb valence fluctuation while the Mg does not. The sentence has been updated to indicate that “intermediate” valence is when there is no clear 3+ and 2+ distinction, so the average valence is something intermediate between the two extremes. More background has been added on the topic.

L50-51 "The low resistance is 50 attributed to either the intermediate valency

of Yb or to the defects in the structure" Electron precise Zintl phase is

generally expected to have lower conductivity. How do these facts coexist?

 Response: In the first paper on Yb14ZnSb11, the low resistivity is attributed to a shift in the Fermi level with addition of  Zn (compared to Al) and slightly contracted lattice. (Suggests that Fermi crosses Yb 4f ). So, basically the band gap is negligibly small because the volume of the unit cell is small. However, defects and intermediate valence of the Yb may also play a role.

L144-151: The statement about 4 crystallographic Yb and Yb2/Yb4 sites is

practically completely duplicated.

Response: The paragraph in question has been revised to eliminate any duplicate statements.

L161: Total occupation of the RE sites exceed 14, while Zn is deficient. Would

this mean RE partially occupies Zn site? No comment is provided

Response: The reported crystal structure of Yb14ZnSb11 presents the Zn site as slightly deficient (96%).We normalized the analysis of the data to 11 Sb in both cases, this make the RE site slightly above 14, but within error (3 sigma), they remain 14, so we decided to keep it that way, but we have added a sentence to explain.

L167-169 The caption contains unrelated infomation regarding PXRD

Response: Unrelated information has been removed.

L171 Please change deficient to deficiency

Response: Deficient has been changed to deficiency.

Figure 3: How do you explain different intensities of the peak around 29.8-30

for blue and red plots?

Response: There is one peak that is not identified and it results from a contribution of a small amount of Yttrium Oxide in the sample.

L210: "Figure 4. (a) Electrical resistivity, (b) thermal conductivity (c) and

Seebeck"

Please change to "Figure 4. (a) Electrical resistivity, (b) thermal conductivity

and (c) Seebeck"

Response: The caption has been changed.

Reviewer 2 Report

The manuscript "Seebeck and Figure of Merit Enhancement by Rare Earth Doping in Yb14-xRExZnSb11 (x=0.5)" relates a study on the thermoelectric/semiconducting characteristics of a unique rare earth system. The results are interesting and provide new insight into how to design and control thermoelectric efficiency while also improving structural characteristics such as melting temperature. The study deals with rare earth elements, so there is a limit in the widespread use of these materials in practical applications, but the fundamental findings are solid and can be applied to future studies and systems. 

The work appears complete and is well written and logical. I dont have anything to suggest adding to the study.

Author Response

The manuscript "Seebeck and Figure of Merit Enhancement by Rare Earth Doping in Yb14-xRExZnSb11 (x=0.5)" relates a study on the thermoelectric/semiconducting characteristics of a unique rare earth system. The results are interesting and provide new insight into how to design and control thermoelectric efficiency while also improving structural characteristics such as melting temperature. The study deals with rare earth elements, so there is a limit in the widespread use of these materials in practical applications, but the fundamental findings are solid and can be applied to future studies and systems. 

The work appears complete and is well written and logical. I dont have anything to suggest adding to the study.

Response: We thank the review for their efforts reading the manuscript and appreciate their comments concerning the quality of the manuscript.